🔓 | **Open Peer Review** | Bacteriophages | Research Article

# Exploration of the genetic landscape of bacterial dsDNA viruses reveals an ANI gap amid extensive mosaicism

Wanangwa Ndovie,[1,2] Jan Havránek,[3] Jade Leconte,[1] Janusz Koszucki,[1,2] Leonid Chindelevitch,[4] Evelien M. Adriaenssens,[5] Rafal J. Mostowy[1]

**ABSTRACT** Average nucleotide identity (ANI) is a widely used metric to estimate genetic relatedness, especially in microbial species delineation. While ANI calculation has been well optimized for bacteria and closely related viral genomes, accurate estimation of ANI below 80%, particularly in large reference data sets, has been challenging due to a lack of accurate and scalable methods. To bridge this gap, we introduce MANIAC, an efficient computational pipeline optimized for estimating ANI and alignment fraction (AF) in viral genomes with divergence around ANI of 70%. Using a rigorous simulation framework, we demonstrate MANIAC's accuracy and scalability compared to existing approaches, even to data sets of hundreds of thousands of viral genomes. Applying MANIAC to a curated data set of complete bacterial dsDNA viruses revealed a multimodal ANI distribution, with a distinct gap around 80%, akin to the bacterial ANI gap (~90%) but shifted, likely due to viral-specific evolutionary processes such as recombination dynamics and mosaicism. We then evaluated ANI and AF as predictors of genus-level taxonomy using a logistic regression model. We found that this model has strong predictive power (PR-AUC = 0.981), but that it works much better for virulent (PR-AUC = 0.997) than temperate (PR-AUC = 0.847) bacterial viruses. This highlights the complexity of taxonomic classification in temperate phages, known for their extensive mosaicism, and cautions against over-reliance on ANI in such cases. MANIAC can be accessed at https://github.com/bioinf-mcb/MANIAC.

**IMPORTANCE** We introduce a novel computational pipeline called MANIAC, designed to accurately assess average nucleotide identity (ANI) and alignment fraction (AF) between diverse viral genomes, scalable to data sets of over 100k genomes. Using computer simulations and real data analyses, we show that MANIAC could accurately estimate genetic relatedness between pairs of viral genomes of around 60%–70% ANI. We applied MANIAC to investigate the question of ANI discontinuity in bacterial dsDNA viruses, finding evidence for an ANI gap, akin to the one seen in bacteria but around ANI of 80%. We then assessed the ability of ANI and AF to predict taxonomic genus boundaries, finding its strong predictive power in virulent, but not in temperate phages. Our results suggest that bacterial dsDNA viruses may exhibit an ANI threshold (on average around 80%) above which recombination helps maintain population cohesiveness, as previously argued in bacteria.

**KEYWORDS** average nucleotide identity, bacteriophage genetics, taxonomy, horizontal gene transfer, alignment fraction, evolutionary biology

Average nucleotide identity (ANI) is one of the most used ways to estimate genetic relatedness between microbial organisms. The classical approach to calculate ANI between a pair of genomes typically involves aligning their nucleotide sequences, either as full genomes or as fragments, and determining the average percentage of

Address correspondence to Rafal J. Mostowy, rafal.mostowy@uj.edu.pl.

The authors declare no conflict of interest.

See the funding table on p. 24.

identical nucleotides across the aligned regions weighted by their lengths (1, 2). While it only approximates evolutionary distance—and only for closely related strains—ANI has played an important role in multiple areas of microbial research, including species delineation and taxonomy (3–11), detecting horizontal gene transfer events (12, 13) and in metagenomics (14–16).

With the advent of next-generation sequencing, the number of microbial genomes available for analysis has increased dramatically, driving the need for faster ANI calculations than those offered by BLAST-based tools like pyani (17). To address this, a variety of tools have been developed to accelerate ANI calculations in various ways. One set of tools relies on a hybrid approach or alignment-based and alignment-free methods. Examples include MUMmer (18), which uses suffix trees to efficiently find exact matches (and was implemented as ANIm in pyani (17), FastANI (7), which uses k-mer sketches and MinHash algorithms to rapidly identify and align homologous regions, or skani (19), which uses a combination of k-mer and sparse approximate alignments). An alternative and competitive approach in terms of speed and accuracy was implemented with the CheckV software (20) based on MegaBLAST—a more efficient implementation of BLAST which employs a greedy algorithm for DNA alignment (21). Finally, there are also tools like Mash (14) and kmer-db (22) which are purely alignment-free and use k-mers to estimate a Mash distance, a proxy for ANI. Despite some loss of accuracy when estimating ANI between distantly related genomes, all of these tools perform well for relatively closely related genomes, particularly in the range of 90%–95% of ANI. The scalability of these tools has therefore facilitated large-scale comparisons on unprecedented scales, providing important data in relation to ongoing debates about the universality of genetic boundaries and a single threshold for species delineation in bacteria (4, 7, 23).

Independently, there has been growing interest in the taxonomic classification of viruses due to the rapid expansion of viral genome databases, with ANI emerging as a key metric for defining species-level boundaries and assessing genomic relatedness (24). Unlike bacterial genomes, viral genomes exhibit greater genetic diversity, presenting unique challenges for taxonomic delineation. To tackle these obstacles, tools like PASC (25), DEmARC (26), Gegenees (27), and SDT (28) were originally developed and used for viral taxonomic classification. However, these tools rely on computationally expensive local or global alignments and do not calculate ANI or alignment fraction (AF) directly, making them less suitable for large-scale genome comparisons. More recently, VIRIDIC has been introduced to calculate both ANI and AF specifically for bacterial viruses (29), aligning with the thresholds recommended by the Bacterial and Archaeal Viruses Subcommittee of the International Committee on Taxonomy of Viruses (ICTV): 95% ANI for species and 70% ANI for genera over their full genome length for dsDNA bacterial and archaeal viruses in the class *Caudoviricetes* (30). However, as VIRIDIC relies on BLAST for accuracy, much like pyani it is computationally intensive and becomes impractically slow for analyzing large reference databases, limiting its use in viral genomics. Despite the advances in both alignment-based and alignment-free ANI calculation tools, a critical gap remains: no method exists that can accurately calculate ANI and AF for highly divergent viral genomes (around 70% ANI) while efficiently scaling to databases of tens or hundreds of thousands of viral genomes. This limitation is particularly significant for bacterial dsDNA viruses, which are both highly mosaic and diverse, emphasizing the importance of the calculation of AF alongside ANI. With the growing number of such genomes being sequenced, it is essential to have a method that can handle this diversity and scale efficiently.

To address this knowledge gap, we here introduce a new approach called MANIAC (MMseqs2-based, ANI Accurate Calculator) that efficiently tackles these challenges. MANIAC is a reproducible, computational pipeline written in SnakeMake using both Python3 and R, that is both fast and sensitive by employing MMseqs2 (31) which filters the DNA sequence space with an alignment-free approach before accurately and efficiently estimating ANI using an alignment-based approach. By tuning its parameters

and using simulated data, we demonstrate that MANIAC reproduces the accuracy of BLAST-based tools like ANIb, and outperforms ANIm, FastANI, CheckV, and skani in predicting ANI for genomes with 60%–70% similarity while being scalable to modern size data sets of viral reference genomes. We then apply MANIAC to two biological problems: investigation of the ANI distribution in bacterial dsDNA viruses and the prediction of taxonomic genus boundaries using ANI therein.

## MATERIALS AND METHODS

### MANIAC workflow

MANIAC is written in Python3 and R, implemented in Snakemake (32). MANIAC runs in three modes: (i) fragment mode, which takes entire genome sequences as input, (ii) coding sequence (CDS) mode, which takes as input nucleotide sequences of predicted genes, commonly obtained using tools like Glimmer or Prodigal, and (iii) protein mode, which uses amino acid sequences for analysis and focuses on calculating amino acid identity (AAI) rather than ANI. The pipeline first pre-processes the results, then performs a search by MMseqs2, and then carries out a calculation of ANI and AF based on these outputs using a combination of several, highly efficient packages in R, including data.table and arrow. The code is accessible at https://github.com/bioinf-mcb/MANIAC.

### ANI calculation

In the fragment mode, MANIAC follows the approach proposed by Goris and colleagues (1). First, genomes are divided into non-overlapping fragments of length $F_L = 1,020$ bp. Next, the search module of MMseqs2 (31) is used to find homologs between these fragments (query) and the full genomes (reference), filtering out all hits that show less that $F_I = 30\%$ identity over the full fragment length and less than $F_C = 0.7$ fragment (query) coverage. MMseqs2 operates through a multi-stage process to efficiently identify similar sequences. During the prefiltering stage, it utilizes a precomputed index table containing k-mers from the reference database, allowing for a rapid comparison against the query sequences. This enables the system to quickly identify potential matches, which are then subjected to a more detailed alignment stage. Once aligned, ANI is then calculated as the mean percentage identity over the length of aligned nucleotides, which is reported separately for when A is compared against B and when B is compared against A. In the post-processing, we calculate a single ANI per pair of A and B by taking the average of the two. In the CDS and protein modes, MANIAC takes as input user-specified CDSs (in nucleotide or amino-acid space, respectively) and uses a best-bidirectional hits approach to approximate orthology (33, 34), after which it applies the same identity-calculation approach. The values of the parameters $F_I$ and $F_C$ in the CDS and protein modes are the same as in the fragment mode.

### *AF calculation*

MANIAC also calculates an alignment fraction (AF) for each pair of genomes. Specifically, for a given alignment of A against B, we calculate the ratio of the total alignment (i.e., sum of the length of all aligned fragments) to the length of A ($\text{AF}_A$) or B ($\text{AF}_B$). In the post-processing step, we provide a single pair of AF values by conservatively taking the minimum alignment length of the two (A vs. B and B vs. A). Based on the single pair of resulting $\text{AF}_A$ and $\text{AF}_B$ values for a pair of genomes A and B, we calculate their mean, minimum, and maximum AF. Unless stated otherwise, we consider only pairs with a minimum alignment length of 500 bp (and 150 aa in the protein mode), with the remaining ones being filtered out and their AF set to 0.

### *Controlling sensitivity and specificity*

In MMseqs2, the k-mer size flag (-k) controls the specificity and sensitivity of the search by specifying the length of the k-mers used in both the query and the precomputed

MMseqs2 index table. For nucleotide sequences, such as those used in MANIAC's fragment and CDS modes, the k-mer size plays a critical role in balancing the accuracy of the results. While a higher k-mer size can enhance specificity by allowing only exact or very close k-mer matches, it may also reduce sensitivity by missing alignments where sequences may be more divergent. Conversely, using shorter k-mers increases sensitivity by identifying a broader range of potential alignments but it also may reduce specificity by incorporating matches that are less precise. This trade-off necessitates careful selection of the k-mer size to achieve an optimal balance between sensitivity and specificity, tailored to the specific requirements of each analysis. The sensitivity flag (-s) controls the prefiltering mode. Higher sensitivity values in this context generate a longer list of similar k-mers, enabling the identification of sequence pairs with lower sequence identity. This increases the ability to find more remote homologs but can also reduce the speed of the search. In nucleotide searches, the sensitivity parameter may have less impact due to the nature of nucleotide matching and the limited nucleotide alphabet, making k-mer length the more critical factor in controlling the search.

### MANIAC parameters

The most important parameters of MANIAC, including k and s, and their values are provided in Table 1. Specifically, --max-seqs specifies the maximum number of sequences aligned with the query and was strongly increased from the default value of 300 to avoid the loss of aligned pairs; --max-seq-len specifies the longest unsplit sequence in the database; -zdrop is the nucleotide drop-off value, similar to xdrop in BLAST; $F_L$ is the fragment length used in the fragment mode; scoring scheme for nucleotide alignment assigns positive scores for matching nucleotides and penalties for mismatches and gaps, helping to quantify the similarity between two sequences based on their aligned regions. We considered three scoring schemes: the standard BLASTN scoring for divergence levels of around 90% (match +3, mismatch −2), UNIT scoring for divergence levels of around 75% (match +1, mismatch −1), and WU scoring for divergence levels of around 65% (match +5, mismatch −4).

The parameter set pyani was chosen to best reflect the parameters defined in the ANIb mode of pyani (17) (see below for the explanation of how pyani estimate ANI and AF). The rationale for the choice of accurate and fast sets of parameters are provided in the Results.

The final output of MANIAC consists of pairs of genomes aligned in either of the three modes (fragment, CDS, and protein). If a pair that was aligned in one mode but was missing in another, it was added to the final output with the zero values of ANI/AAI and AF. This approach ensures the inclusion of pairs with any degree of relatedness identified in at least one mode, while pairs not aligned in any mode are excluded from the final output to manage the data size effectively.

**TABLE 1** Parameter values for different parameter sets of MANIAC, where "ANIb" refers to the baseline comparison to pyani's ANIb mode, while "Accurate" and "Fast" are the sets obtained via the parameter optimization in Results

| Parameter set | ANIb | Accurate | Fast |
| --- | --- | --- | --- |
| max-seqs | 20,000 | 20,000 | 20,000 |
| max-seq-len | 100,000 | 65,000 | 65,000 |
| zdrop | 150 | 40 | 40 |
| s | 7.5 | 7.5 | 7.5 |
| k | 11 | 11 | 15 |
| $F_L$ | 1,020 | 500 | 1,020 |
| scoring scheme | BLASTN | UNIT | UNIT |

## Estimating ANI and/or AF using other approaches

### ANIb (pyani with BLAST)

As the main benchmark, we calculated ANI and AF using the ANIb method implemented in pyani (17). The ANIb method fragments one genome (query) in pieces of 1,020 bp and compares them to a target genome using BLAST+. For each comparison between genomes, this method calculates ANI following the approach of Goris et al. (1), namely, as the percentage identity of the aligned fragments of at least 30% identity over the full fragment length and at least 0.7 fragment coverage; it also calculates alignment coverage (fraction of query that is aligned). The average of query coverage and subject coverage returned by pyani was directly comparable to our definition of alignment fraction (AF). Unless stated otherwise, the results were filtered by a minimum alignment length of 500.

### ANIm (pyani with MUMmer)

MUMmer is a tool that efficiently aligns DNA and amino acid sequences using a suffix tree data structure for fast pattern matching (18). An implementation of ANI calculation using MUMmer is provided also in pyani (17), and is referred to as ANIm. Both ANI and AF are calculated analogously as with ANIb, with the difference that the query is aligned against the target genome using NUCmer, a component of MUMmer.

### FastANI

FastANI is an alignment-free tool that calculates ANI and genome coverage (35). For a given pair of genomes, FastANI fragments the query genome into non-overlapping fragments and then maps them onto the reference genome using mashmap. Spatially proximate mappings are grouped into bins, and ANI is determined by calculating the weighted mean of the average nucleotide identities across these bins. The coverage for a genome pair using FastANI is quantified as the proportion of query genome fragments that successfully map to the reference genome, and hence directly comparable to MANIAC's estimate of AF. We ran FastANI (version 1.33) with the following parameters: k = 15 (following the recommended length for Mash in viruses (36), fragLen = 500 (based on the fragment length analysis by (37), and minFraction = 0.1 (to capture pairs with low AF but avoiding noise stemming from short alignments). The ANI of each pair was then taken as the mean of the query-reference ANI and query/fragment AF was reported as the coverage calculated by FastANI.

### CheckV

CheckV is a pipeline designed to identify closed viral genomes, estimate genome fragment completeness, and remove host regions from integrated proviruses (20). It includes a script to calculate ANI and AF values for pairs of genomes based on a table of local alignments generated via MegaBLAST (38). Then, ANI is calculated by taking a weighted average of the percentage identities across all alignment segments for each query-target pair, while AF is calculated by merging overlapping aligned regions for both the query and target, and determining what proportion of each sequence is aligned to the other. For each genome pair, ANI was averaged in both directions (A to B and B to A) for consistency.

### skani

skani is an alignment-free tool that estimates ANI and AF between two genomes by identifying matching DNA regions without performing a detailed alignment (19). It begins by prefiltering genomes using a sparse set of k-mers, quickly filtering out genomes with an estimated ANI below 80%. For genomes that pass this threshold, skani fragments the query genome into 20 kb chunks and compares these chunks to the reference genome. Within each chunk, it identifies chains of matching k-mers,

representing likely homologous regions, and calculates ANI as the proportion of identical bases within these matches. AF is then calculated as the fraction of each genome covered by all matching regions. As recommended by the authors, we (i) used the triangle mode for all-by-all comparisons, (ii) set the -m parameter to 200 (optimized for small contigs), and enabled the --medium setting to enhance accuracy for genomes with lower ANI at a modest cost to speed. The final ANI for a pair (A, B) is the average of ANI from A to B and B to A.

## Data used for benchmarking

### Genomic data

In October 2022, we downloaded a comprehensive data set consisting of 22,774 phage genomes, along with associated metadata, from the Inphared database (39). Genes were predicted with prodigal using the -p meta option (40). To refine the data set for accuracy and relevance, we incorporated the "exclusion list" from the most recent Inphared update as of March 2024, resulting in the exclusion of certain isolates and a revised count of 22,616 genomes. Of those, 4,655 were part of RefSeq and are henceforth referred to as the RefSeq data set; 500 of those genomes were randomly selected to benchmark the accuracy and efficiency of MANIAC against established methods such as BLAST (ANIb), MUMmer (ANIm), FastANI, CheckV, and skani (the RefSeq 500 data set). We also downloaded taxonomic information from the Virus Metadata Resource from the International Committee on Taxonomy of Viruses (ICTV), release VMR_MSL39_v2 (henceforth called ICTV 39) (41). We then employed CheckV (20) to evaluate the completeness of the Inphared genomes, setting a threshold for inclusion at greater than 90% for those that are absent in the ICTV 39 or RefSeq. This resulted in a curated data set of 21,780 high-quality phage genomes, among which 8,605 are present in ICTV 39. In addition, we constructed a distinct taxonomic data set from ICTV 39 of bacterial dsDNA viruses, removing any isolates not present in the Inphared database or those present on the exclusion list. In cases of isolates appearing in both RefSeq and GenBank, we identified and eliminated segmentations and duplications to avoid redundancy. This approach yielded a refined data set of 4,618 unique species of bacterial dsDNA viruses, henceforth referred to as the ICTV data set, which was used for machine learning of the taxonomic same-genus predictions. To test the scalability of MANIAC to larger viral data sets, we a previously published database of 142,809 non-redundant gut phage genomes from 28,060 metagenomes called the Gut Phage Database (42) (the GPD data set), and the PhageScope database from which we managed to download 873,717 phage genomes (42) (the PhageScope data set)

### Simulated data

To simulate true values of ANI and AF, we developed an approach to generate genome-like string pairs with predefined evolutionary distance $d$ and coverage $c$. Coverage ($c$) was defined as the fraction of orthologous gene-like strings relative to the total number of genes in each genome with $c = N_O/N$, where $N_O$ represents the number of orthologous genes, and $N$ is the total number of genes in the genome such that $N = N_O + N_U$, where $N_U$ denotes the number of unrelated genes (by default we assume $N = 75$). Gene lengths were simulated from a log-normal distribution, with a mean of 2.22 and a standard deviation of 0.34 in base-10 logarithms, and multiplied by 3 to represent nucleotide sequence lengths. Since gene lengths varied between genome pairs, the actual AF for each genome pair was calculated based on the total length of orthologous genes relative to the total genome length. On average, the AF converged to the predefined coverage $c$, but variations in gene lengths introduced fluctuations around this average.

To simulate genome sequences, orthologous genes were concatenated in the same order for both genomes, while unrelated genes were generated as independent random DNA strings. Orthologous genes were evolved with a predefined evolutionary distance

$d$ using Seq-Gen and the HKY model (43), and ANI was calculated by determining the proportion of identical nucleotides in the orthologous gene sequences. Unrelated genes were divided into blocks using a multinomial distribution, with an average block size of eight genes. These unrelated gene blocks were then randomly inserted into the genomes to create mosaic structures, contributing to the alignment fraction. To simulate deviations from synteny, we shuffled a proportion $s$ of the orthologous genes in one genome, where $s$ varied between 0 and 1 (we assumed $s = 0$ unless stated otherwise). For each combination of evolutionary distance $d$ and coverage $c$, we simulated 20 genome pairs. We varied the evolutionary distance $d$ from 0 to 0.8 (corresponding to ANI between 1 and 0.5), and $c$ from 1 to 0.1, in increments of 0.05, with 20 genome pairs simulated for each combination of $d$ and $c$, yielding a total of 6,840 simulated genome pairs (13,680 simulated genomes).

## Measuring the accuracy of prediction and runtime

We calculated the root mean squared error (RMSE) as the mean difference between the predicted and actual values of ANI and AF values. When assessing RMSE on real data, the actual value was the one estimated by ANIb. Confidence intervals for linear regression were calculated by performing 1,000 bootstrap iterations to estimate the coefficients. We then calculated the median and 95% quantile range for these coefficients. To estimate the run times for different ANI estimation tools calculations, we executed all commands on a High-Performance Cluster, with 192 cores, 300 GB of RAM with 72 h of the run time limit, and used the time command to measure the actual run times of each program.

## Lifestyle and taxonomic prediction

We used BACPHLIP v0.9.6 to predict the phage lifestyle (44). Phages were assigned as virulent if their score was 0.95 and greater and as temperate if their score was less than 0.05, yielding 6,496 isolates predicted as virulent, 4,931 as temperate, and 10,353 without lifestyle prediction. We used the Virus Metadata Resource from the International Committee on Taxonomy of Viruses (ICTV) to assign genus-level taxonomy to the NCBI genomes for the releases 39 (VMR_MSL39_v2), 38 (VMR_MSL38_v2), 37 (VMR_21–221122_MSL37), 36 (VMR_18–191021_MSL36), and 35 (VMR_15–010820_MSL35) from https://ictv.global/vmr.

## Same-genus classification with machine learning

We employed a logistic regression approach to classify pairs of phage genomes as belonging to the same or different genera based on their ANI and AF values. To this end, we used the "ICTV data set", which consisted of 4,617 representative genomes of bacterial dsDNA virus species with a known genus. Logistic regression was chosen for its simplicity, interpretability, and effectiveness in binary classification tasks, making it well-suited for analyzing the relationship between genomic similarity metrics (ANI and AF) and taxonomic classification.

### Model description

The logistic regression model was formulated with ANI (or AAI) and AF as the independent variables and the binary outcome variable indicating whether a pair of genomes belong to the same genus (TRUE) or not (FALSE). The model estimates the probability of the TRUE outcome based on their ANI (or AAI) and AF values, using a logistic function to map linear combinations of the input variables to probabilities.

### Data splitting and cross-validation

To ensure a rigorous assessment of model performance, the ICTV data set comprising pairs of phage genomes was initially divided into two distinct sets: (i) a combined

training/validation set and (ii) a separate test set. This division was executed at the genus level, not at the level of genome pairs, adhering to a ratio of 65% for training/validation and 35% for the test set. The random seed was chosen such that the same ratio on the level of genome pairs was approximately 65%/35% too. For the process of model optimization and parameter tuning, we exclusively utilized the training/validation set, implementing a 10-fold cross-validation strategy. Specifically, this set was further divided into 10 equal parts, or "folds." We then stratified the cross-validation process by genus pairs, hence ensuring that pairs of viral genera belonging to specific folds were grouped consistently, preventing the same genus pairs from appearing in both the training and validation sets within a single fold. During each iteration of cross-validation, nine of these folds were used for training the model, while the remaining single fold served as a validation set to assess model performance. This cycle was repeated ten times, ensuring that each fold acted as the validation set once, thereby allowing for a comprehensive internal evaluation of the model's predictive capabilities. It is important to note that the results derived from this 10-fold cross-validation process pertain solely to the training/validation phase, aimed at refining our models within a controlled subset of the data. By contrast, the independent test set was reserved for a final, unbiased evaluation of the model's performance. The metrics reported for the test set, therefore, reflect the model's ability to generalize to new, unseen data, distinct from the internal validation achieved through cross-validation.

### Model evaluation

For model evaluation, we focused on precision-recall metrics, given the class imbalance typically observed in genomic data sets. In the context of our study, precision (the proportion of true positive predictions among all positive predictions) and recall (the proportion of true positive predictions among all actual positives) are particularly informative. Specifically:

- Precision is defined as follows:

$$\text{precision} = \frac{\text{true positives (TP)}}{\text{true positives (TP)} + \text{false positives (FP)}}$$

Precision is critical for ensuring that when pairs are classified as belonging to the same genus, this classification is likely correct, minimizing false positives.

- Recall is defined as follows:

$$\text{recall} = \frac{\text{true positives (TP)}}{\text{true positives (TP)} + \text{false negatives (FN)}}$$

Recall assesses the model's ability to identify all relevant pairs that truly belong to the same genus, minimizing false negatives.

The choice of the precision-recall (PR) curve and area under the PR curve (PR-AUC) as metrics for model performance is justified by the imbalanced nature of our data set. This imbalance is evident, as the number of pairs belonging to the same genus is significantly lower than the number of pairs from different genera, rendering accuracy or the traditional ROC-AUC metric less suitable for our analysis. Both precision and recall take into consideration only cases where either the true class or the predicted class is positive, and neither metric relies on the true negative pairs, which comprise the majority of pairs for which the ANI estimation failed. In addition, we also reported on the precision, recall, false discovery rate or FDR, defined as $1 - \text{precision}$, and F1 score, defined as

$$\text{F1} = 2 \times \frac{\text{precision} \times \text{recall}}{\text{precision} + \text{recall}}$$

The uncertainty in the PR-AUC was estimated using bootstrap resampling (100 iterations), where predictions and true labels were randomly sampled with replacement. The PR-AUC scores derived from these bootstrap samples were then used to calculate the 95% confidence interval.

## RESULTS

### MANIAC achieves near-perfect correlation with ANIb in ANI and AF estimation

To evaluate MANIAC's performance, we benchmarked its estimates of ANI and AF against the ANIb mode of pyani, an implementation of the BLAST-based Goris et al. approach (1), assuming it to be the known "truth" (see Materials and Methods for details). We parametrized MANIAC using the parameters shown as the "pyani parameter setting" shown in Table 1. Our analysis employed a data set comprising 500 randomly sampled phage genomes from the NCBI RefSeq database (RefSeq 500 data set; see Materials and Methods). In addition, we compared MANIAC's results to those from ANIm, FastANI, CheckV, and skani (see Materials and Methods). As illustrated in Fig. 1 and Fig. S1, MANIAC achieved a near-perfect correlation of $R^2 \approx 1$ with ANIb in estimating both ANI (Fig. 1A) and AF (Fig. 1B), closely approximating the ideal $Y = X$ relationship. Specifically, the summary of linear regression parameters, $R^2$, and 95% confidence intervals (CI) for each method's correlation with ANIb are provided in Table 2. MANIAC outperformed all other methods, showing the highest $R^2$ and coefficients closest to the ideal linear relationship for both ANI and AF.

Compared to MANIAC, all other tested methods displayed lower correlations with ANIb (Table 2), deviating significantly from the ideal $Y = X$ relationship and exhibiting tendencies to over- or under-estimate ANI, particularly for more distantly related genome pairs. Likewise, AF estimates by FastANI, MUMmer, skani, and CheckV showed poorer performance relative to those of MANIAC (Fig. 1B). We hypothesized that the inferior performance of other tools may be due to the noise introduced by the short alignment length (even though it was originally filtered at 2,000 bp). Hence, we reexamined the correlation values between ANIb and other methods only for the genomes with a minimum AF > 0.5. As shown in Fig. 1 and summarized in Table 2, while correlations improved for all methods, MANIAC remained the only method with a correlation approximating the $Y = X$, with CheckV being the close second. Overall, MANIAC is an excellent alternative to ANIb, delivering comparable accuracy with much faster computation, though it trades some speed for higher accuracy compared to other tools (Fig. S1).

### Optimization of MANIAC using simulated genome pairs

While MANIAC demonstrated high accuracy in comparison to ANIb, this approach is limited as ANIb is not a known "ground truth" and requires optimization, particularly for viral genomes. To conduct a more rigorous assessment, we generated a simulated data set with pre-defined ANI and AF values, allowing us to compare MANIAC's estimates directly against a true benchmark. This approach ensures that our comparisons are unbiased and tailored to reflect viral genome properties, such as high sequence divergence and genetic mosaicism. We used simulations that generated pairs of synthetic genomes with predefined evolutionary distance $d$ and coverage values $c$, where $c$ is defined as the proportion of orthologous genes to all genes (see Materials and Methods). For each pair, we calculated the true ANI as the proportion of identical nucleotides in orthologous genes, while the AF was defined by the total length of orthologous genes relative to genome length (hence, we defined AF as coverage weighted by gene lengths). This approach provided us with a reliable simulated "ground truth" that could be leveraged to provide a much less biased benchmark for different approaches to estimate ANI and AF.

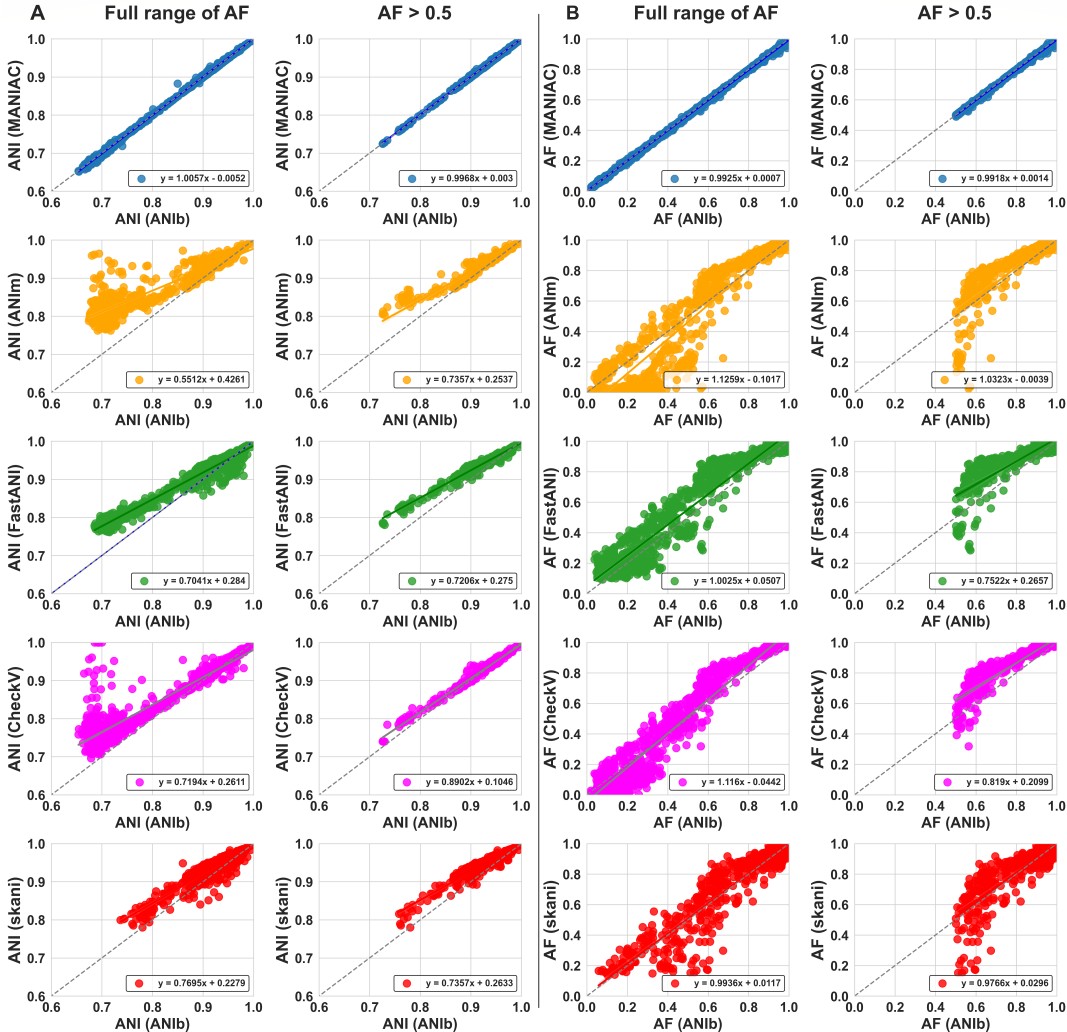

**FIG 1** Comparison of estimated ANI and AF values against ANIb. (A) ANI estimated by ANIb (*X*-axis) vs. ANI estimated by MANIAC (blue), ANIm (orange), FastANI (orange), CheckV (gray), and skani (red) obtained for the RefSeq500 data set (see Materials and Methods). The left column shows the full range of AF, right column shows pairs with AF ≥ 0.5 estimated by ANIb. (B) AF estimated by ANIb (*X*-axis) vs. AF estimated by the remaining methods. The color code and left/right column are analogous to panel A. The dashed line shows the $Y = X$ relationship. The results of pyani and MANIAC were filtered by a minimum alignment length of 2,000 bp.

To achieve an optimized version of MANIAC, we evaluated a series of key parameters using simulated genome pairs. Specifically, we started with the ANIb set of parameters used in Fig. 1 and varied them while investigating their impact on the RMSE assessment metric in different ranges of the evolutionary distance $d$. The first parameter was the fragment size $F_L$, which is only used in the fragment mode of MANIAC and determines the length of chopped genome fragments as proposed by Goris et al. (1). We found that the best results are achieved by $F_L = 500$ bp (Fig. S2) because longer fragments may combine mosaic fragments which bias the ANI/AF estimates while short fragments are less likely to align when distant. The optimal value of $F_L = 500$ bp provides an optimum and was set in the subsequent assessments.

Next, we investigated the impact of the k-mer length and the scoring scheme for nucleotide alignments on the ANI estimation (Fig. S3). We hypothesized that the scoring scheme could affect such estimation because it is intrinsically linked to the expected level of evolutionary divergence, influencing the algorithm's sensitivity to detect related sequences and its ability to accurately estimate ANI across different divergence ranges (45, 46). Specifically, BLASTN scoring is optimized for detecting sequences with

**TABLE 2** Summary statistics for ANI and AF estimation when using ANIb as the benchmark as shown in Fig. 1[a]

| Metric | Method | R-squared | R-squared 95% CI | Intercept | Intercept 95% CI | Coefficient | Coefficient 95% CI |
|---|---|---|---|---|---|---|---|
| ANI estimation (full range of AF) | MANIAC | 0.9995 | (0.9993, 0.9996) | −0.0052 | (−0.006,−0.0044) | 1.0057 | (1.0047, 1.0066) |
| | ANIm | 0.854 | (0.8213, 0.8829) | 0.4262 | (0.4126, 0.4404) | 0.5512 | (0.535, 0.5664) |
| | FastANI | 0.9637 | (0.9571, 0.9697) | 0.284 | (0.2776, 0.291) | 0.7041 | (0.6957, 0.7119) |
| | CheckV | 0.8776 | (0.8418, 0.9102) | 0.2611 | (0.248, 0.2742) | 0.7194 | (0.7047, 0.734) |
| | Skani | 0.8625 | (0.8305, 0.8898) | 0.2279 | (0.2075, 0.2492) | 0.7695 | (0.7467, 0.7918) |
| ANI estimation (AF≥ 50%) | MANIAC | 0.9992 | (0.9989, 0.9994) | 0.003 | (0.0009, 0.0053) | 0.9968 | (0.9944, 0.9991) |
| | ANIm | 0.9203 | (0.9045, 0.9338) | 0.2537 | (0.2181, 0.2809) | 0.7357 | (0.7061, 0.7731) |
| | FastANI | 0.9615 | (0.9523, 0.9693) | 0.275 | (0.2627, 0.2898) | 0.7206 | (0.7045, 0.7341) |
| | CheckV | 0.9854 | (0.9809, 0.9893) | 0.1046 | (0.0923, 0.1176) | 0.8902 | (0.8763, 0.9033) |
| | Skani | 0.9143 | (0.8945, 0.9309) | 0.2633 | (0.2366, 0.2922) | 0.7357 | (0.7052, 0.7644) |
| AF estimation (full range of AF) | MANIAC | 0.9996 | (0.9996, 0.9997) | 0.0007 | (0.0003, 0.0011) | 0.9925 | (0.9913, 0.9937) |
| | ANIm | 0.9055 | (0.894, 0.9166) | −0.1017 | (−0.114,−0.0894) | 1.1259 | (1.1117, 1.1401) |
| | FastANI | 0.9214 | (0.9132, 0.9294) | 0.0507 | (0.0413, 0.061) | 1.0025 | (0.9899, 1.0166) |
| | CheckV | 0.9608 | (0.957, 0.9646) | −0.0442 | (−0.0496,−0.0396) | 1.116 | (1.1085, 1.1241) |
| | Skani | 0.8681 | (0.8451, 0.8893) | 0.0117 | (−0.0128, 0.0328) | 0.9936 | (0.9702, 1.0201) |
| AF estimation (AF≥ 50%) | MANIAC | 0.9968 | (0.9961, 0.9975) | 0.0014 | (−0.0027, 0.0052) | 0.9918 | (0.9869, 0.9972) |
| | ANIm | 0.7175 | (0.681, 0.7577) | −0.0039 | (−0.0794, 0.0791) | 1.0323 | (0.9407, 1.1168) |
| | FastANI | 0.7084 | (0.6728, 0.7492) | 0.2657 | (0.2092, 0.3205) | 0.7522 | (0.691, 0.8149) |
| | CheckV | 0.8441 | (0.8178, 0.8698) | 0.2099 | (0.1694, 0.2519) | 0.819 | (0.7727, 0.8638) |
| | Skani | 0.708 | (0.6765, 0.747) | 0.0296 | (−0.037, 0.0984) | 0.9766 | (0.8995, 1.0509) |

[a]R-squared and linear regression parameters (intercept, coefficient) are presented with 95% confidence intervals (CIs) and estimated by bootstrap (see Materials and Methods). MANIAC's parameters used are provided in Table 1 (ANIb parameter set).

divergence levels around ANI ~90%, making it well-suited for bacterial genomes, while the UNIT scoring scheme better reflects the evolutionary divergence at ANI ~70%, which is more relevant for levels of divergence observed in viruses. Results demonstrate that, as expected, the estimates improve with a decreasing value of $k$ gave the best results with $k = 11$ being the close second. We also found that the UNIT scoring is much better at capturing the levels of divergence around ANI ~70% (important for taxonomic classification in many viruses) than BLASTN, while the results of UNIT and WU are indistinguishable. (Henceforth, we omit the results for the WU scheme since they are quantitatively identical to those of the UNIT scheme.) The impact of the parameters zdrop and max-seq-len on the ANI/AF estimation is shown in Fig. S4 and S5, respectively. The results point to a negligible impact of these parameters on RMSE as long as zdrop is at least 40.

Finally, we used the NCBI RefSeq data set to measure the runtimes depending on the values of the above parameters (Fig. S6). The results point to a trade-off between accuracy and runtime for the value of $k$, suggesting that $k = 11$ is the optimal choice when accuracy is the priority at sensible speed, while $k = 15$ is the better choice when speed is the priority at sensible accuracy. There is also a trade-off in the fragment length $F_L$ parameter as larger fragments decrease both the runtime and RMSE, again pointing to $F_L = 500$ as the better choice for accuracy and $F_L = 1,020$ being a better choice for speed. Given a negligible impact of zdrop and max-seq-len on accuracy but a visible one on runtime, we decided to keep those values as the default values of mmseqs, namely zdrop of 40 and max-seq-len of 65,000. Finally, the UNIT scoring scheme turned out to be the best choice, both in terms of accuracy and runtime. Thus, as shown in Table 1, henceforth we use MANIAC in two parameter sets: Fast ($k = 15$ and $F_L = 1,020$) and Accurate ($k = 11$ and $F_L = 500$).

## MANIAC accurately estimates relatedness for simulated genome pairs

Having optimized MANIAC using simulated data, we next assessed the performance of MANIAC in fragment mode (Accurate and Fast parameter sets) and the CDS mode compared to four other tools: FastANI, MUMmer, CheckV, and skani. The results, shown in Fig. 2, highlight several key findings from this benchmarking comparison. First, MANIAC outperformed all other tools in estimating ANI, especially at large evolutionary distances of 0.4 to 0.5 (corresponding to ANI around 60%–70%), a range where other tools struggled (Fig. 2A). This high accuracy was consistent across all three MANIAC configurations (Accurate, Fast, and CDS) and was maintained regardless of the coverage values used in the simulations. Second, MANIAC Fast achieved ANI estimation performance comparable to MANIAC Accurate, demonstrating that the faster configuration does not significantly compromise accuracy in ANI calculations. Third, MANIAC proved superior to all other tools in terms of AF estimation, particularly for more distantly related genome pairs (Fig. 2B and C). While CheckV, which relies on full genome alignment rather than the Goris et al. approach, showed high accuracy in AF estimation for closely related genomes, its performance declined with increased evolutionary divergence (see also Discussion on the pros and cons of different ways of AF calculation). This limitation gave MANIAC a distinct advantage in accurately estimating AF for more diverse genome pairs (Fig. 2C). Finally, MANIAC in CDS mode was the best overall for both ANI and AF estimation; however, this mode requires an additional step of ORF prediction, adding computational cost. In summary, MANIAC offers an accurate alternative for calculating ANI and AF, particularly suited to the complexities of divergent and mosaic genomes, such as those found in viral genomes.

## MANIAC is scalable to large data sets

To gauge the scalability of MANIAC to large data sets, we assessed its runtime in two parameter settings (Accurate, Fast; see Table 1) for four datasets of increasing size: the Refseq data set, the Inphared data set, the GPD data set, and the PhageScope data set (see Materials and Methods), comparing it directly to the runtimes of the other methods used here (ANIm, FastANI, CheckV, and skani). The results, shown in Fig. 3, show that

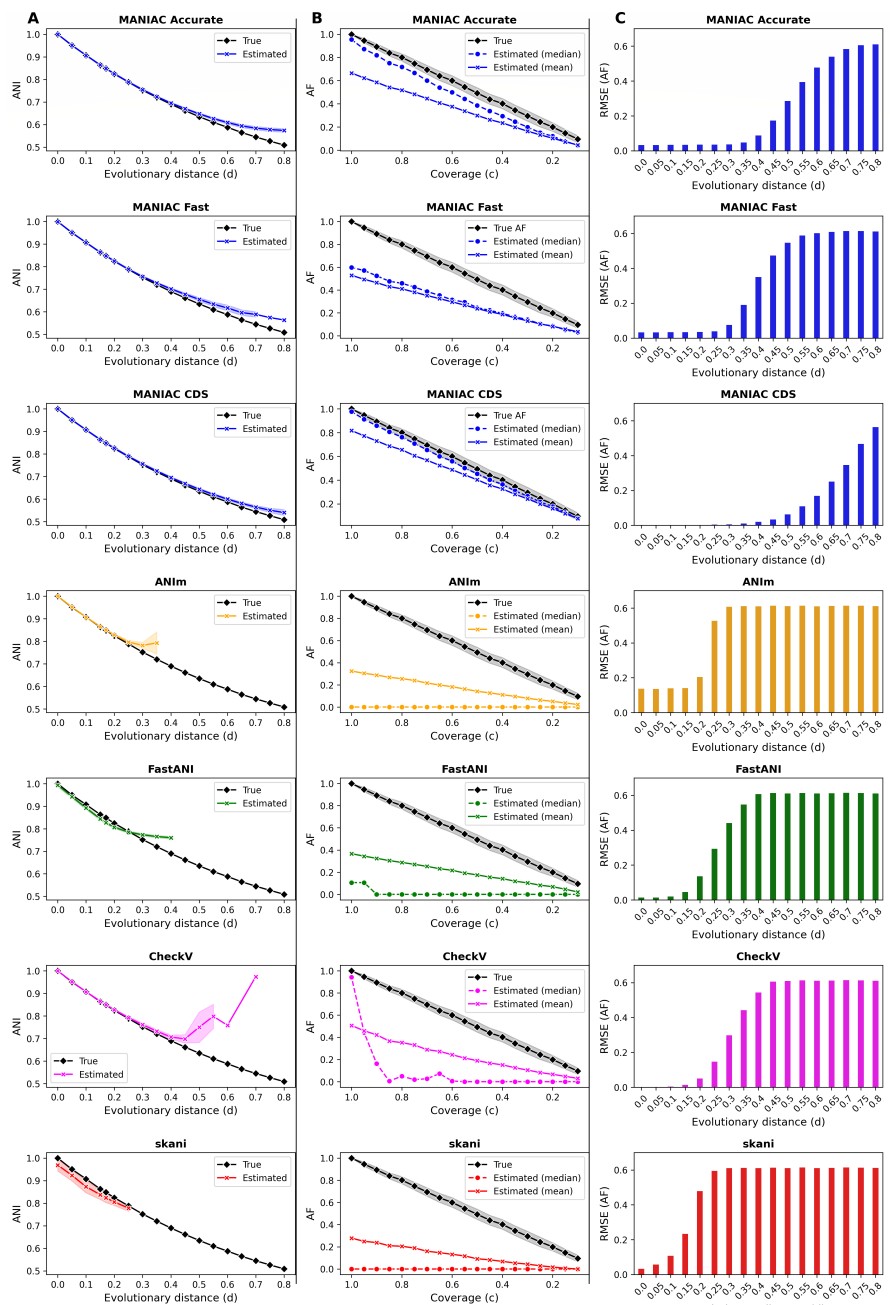

**FIG 2** Comparison of estimated ANI and AF values against simulated data. (A) Simulated evolutionary distance $d$ (*X*-axis) vs. true ANI (black diamond) and estimated ANI (cross). Different rows show estimation by various tools: MANIAC Accurate, MANIAC Fast, MANIAC CDS (all blue), FastANI (green), MUMmer (orange), CheckV (magenta), and skani (red). (B) Simulated coverage $c$ versus true AF (black diamond) and estimated AF value by mean (cross) and median (circle). Large differences between the mean and median values are due to the highly skewed distribution of estimated AF values driven by distant genome pairs. (C) RMSE of estimate AF as a function of evolutionary distance $d$, with colors and methods the same as in panels A and B. Parameters used for MANIAC CDS are the same as in the Accurate set.

skani is the fastest method, though it achieves this speed at the cost of accuracy (cf., Fig. 2), particularly for more distantly related genomes. ANIm, on the other hand, did not complete a single of the data sets due to exceeding the allocated 300 GB of RAM memory. Among the remaining approaches, FastANI was faster than CheckV or MANIAC, but it also ran out of memory on the PhageScope data set. The runtime of CheckV

was near-identical to MANIAC, but it did not complete on the PhageScope data set either, hitting a wall time of 72 h. MANIAC-Accurate was the slowest of the tools (except for ANIm) but did complete the GPD data set. Finally, MANIAC-Fast was the only tool (except for skani) that completed the PhageScope data set. These results suggest that MANIAC offers the best approach for accurate ANI estimation in high-quality reference databases of viral genomes, especially when determining relationships at ANI levels below 95% is critical. Moreover, MANIAC is well-suited for large data sets, even those of metagenomic scale with over 100,000 genomes, making it a robust and scalable option for comprehensive ANI analysis.

## Validating simulation predictions with real genomic data

The main prediction of the simulation approach is that the choice of the scoring scheme and MANIAC's operational mode (fragment vs. CDS) can affect the bias in estimates of ANI and AF for more distantly related pairs of genomes (cf., Fig. 2; Fig. S3). Before proceeding to apply MANIAC to specific biological questions, we tested whether these types of bias can be reproduced when using average amino acid identity (AAI) as a more sensitive approach to estimate genetic relatedness between viral genomes. To this end, we executed MANIAC on the RefSeq data set in all three modes (fragment, CDS, and protein). We employed the Accurate parameter setting for both the fragment and CDS modes. In the protein mode, which calculates AAI across all genome pairs, we correlated these AAI values with ANI values derived from the fragment and CDS modes. To ensure robustness in our ANI/AAI comparisons, we only included genome pairs with substantial coverage (total alignment length exceeding 5,000 amino acids).

Confirming our simulation-based predictions, the scoring scheme significantly impacted the ANI estimations for genome pairs at the borderline of the taxonomic genus range (ANI of 70% and below). Fig 4A illustrates a strong correlation between ANI and AAI estimates; however, the relation is notably different between the BLASTN and UNIT scoring schemes for distant genome pairs, especially for ANI values below 70% (Fig. 4A and B). The disparity in ANI estimates between these two scoring schemes could surpass five percentage points for genome pairs with an AAI less than 80%, which could affect ANI-based taxonomic assignment in many viruses.

Also in line with our simulation findings, AF estimation in the fragment mode was found to be less reliable for more distantly related genomes. This underestimation of AF, as evidenced by comparisons with the protein mode (considered here as "true" AF), became more pronounced with increasing genomic distance (Fig. 4C and D). The fact that the underestimation of AF, when measured in this way, occurs also for closely related genome pairs suggests that deviation from synteny (via rampant genetic mosaicism) is one important factor contributing to this effect (the effect also visible in our simulations, see Fig. S7), but the failure to align fragments is also very likely to contribute as ANI gets close to the twilight zone of 0.5. Finally, consistent with the results shown in Fig. 2, we found that the accuracy of AF estimation, when using the protein mode as the benchmark, was highest in the CDS mode but comparable to the fragment mode for the Accurate parameter setting (see Fig. S8).

In conclusion, through the integration of both simulated and real data sets, we demonstrate that MANIAC, when accurately parameterized, serves as the optimal method for assessing genetic relatedness for diverse genome pairs in reference genome data sets, even when exceeding 100 k genomes. We thus next employed it to investigate the genetic landscape of bacterial dsDNA viruses and their taxonomic assignment.

## ANI distribution in dsDNA bacterial viruses is multimodal

After calibrating and validating MANIAC through both simulated and genomic data sets, we next applied the pairwise calculation of ANI and AF between phage genomes to two biological problems, the first of which is the investigation of the ANI distribution in phage populations.

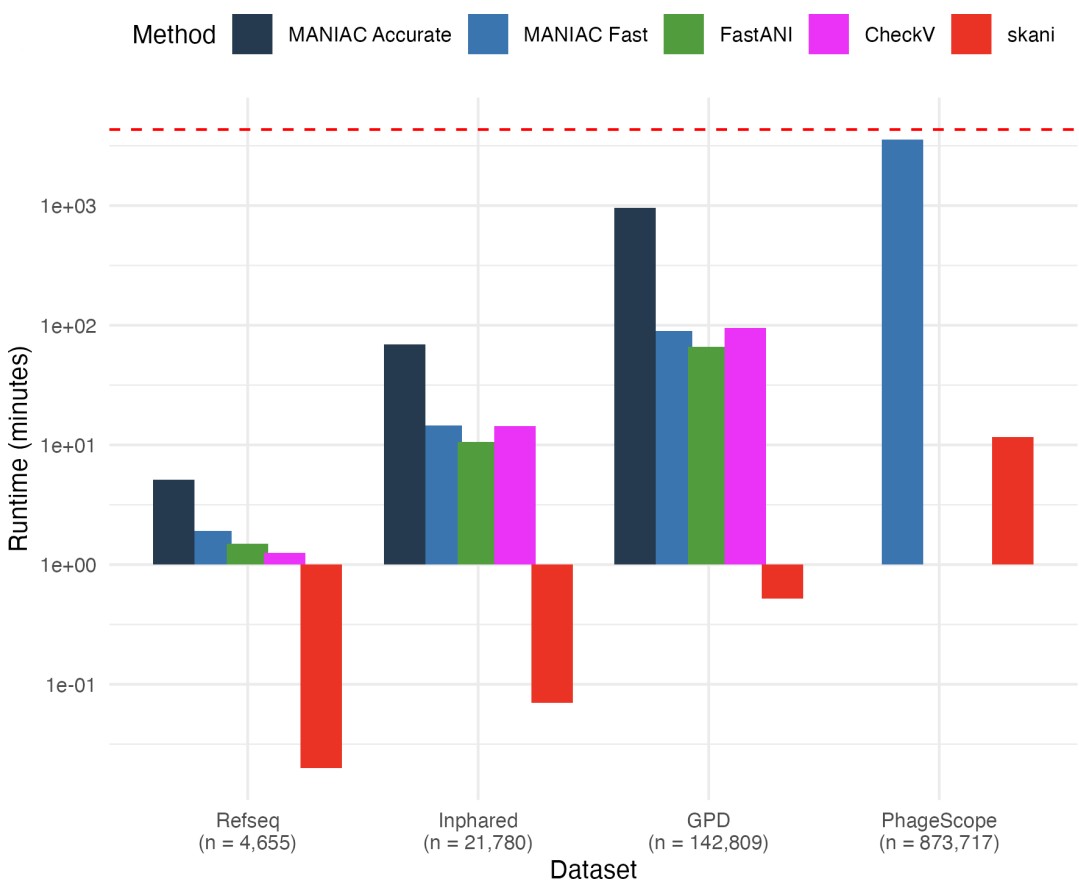

**FIG 3** Scalability of MANIAC. Runtimes in minutes (*Y*-axis) of MANIAC Accurate (dark blue), MANIAC Fast (light blue), FastANI (green), CheckV (magenta), and skani (red) on four different data sets (*X*-axis): the RefSeq data set, the Inphared data set, the GPD data set, and the PhageScope data set using a Linux machine with 192 threads and 300 GB of memory. ANIm is not shown as it did not finish on any of the data sets. PhageScope is missing the results of MANIAC Accurate (exceeded wall time of 72 h), CheckV (same reason), and FastANI (exceeded the allocated memory). The red dashed line denotes the wall time of 72 h.

Large-scale comparisons of microbial genomes have revealed that bacterial and archaeal populations exhibit a distribution of genomic sequence similarity, as measured by ANI, which features a distinct gap—commonly referred to as the "ANI gap" (7, 47, 48). In bacteria, this gap typically occurs in the range of 85%–95%, with most intra-species pairs displaying a high similarity of ANI ≥ 95%. The existence of a similar ANI gap in bacterial viruses has been the subject of an ongoing debate. For example, while genomic clusters of diverse dsDNA lytic cyanophages at 95% ANI form ecologically meaningful units based on host range (49), it has also been observed that sequence thresholds may not consistently define meaningful populations across viral groups due to highly variable recombination rates (50). To account for this variability, a consensus was proposed to apply the 95% ANI threshold to define "viral operational taxonomic units," or vOTUs, rather than species (51). In addition, a discontinuity of around 99.5% ANI has been noted in phages and other viruses, reflecting intra-species populations known as "genovars" (52). Finally, a study by Acetto and Janez also reported a discontinuity of around 80%–90% ANI in a data set of 186 lytic myoviruses (53); however, data sets of this size are inherently prone to sampling bias, making it difficult to generalize findings across bacterial dsDNA viruses. To address this limitation, we applied MANIAC to two datasets: the Inphared data set, which includes 21,754 complete genomes of bacterial viruses, of which 18,583 have been assigned to the *Caudoviricetes* class, and the ICTV data set, which consists of 4,617 genomes bacterial dsDNA viruses, of which 99.2% have been assigned to the *Caudoviricetes* class. For both data sets, we calculated ANI in the fragment

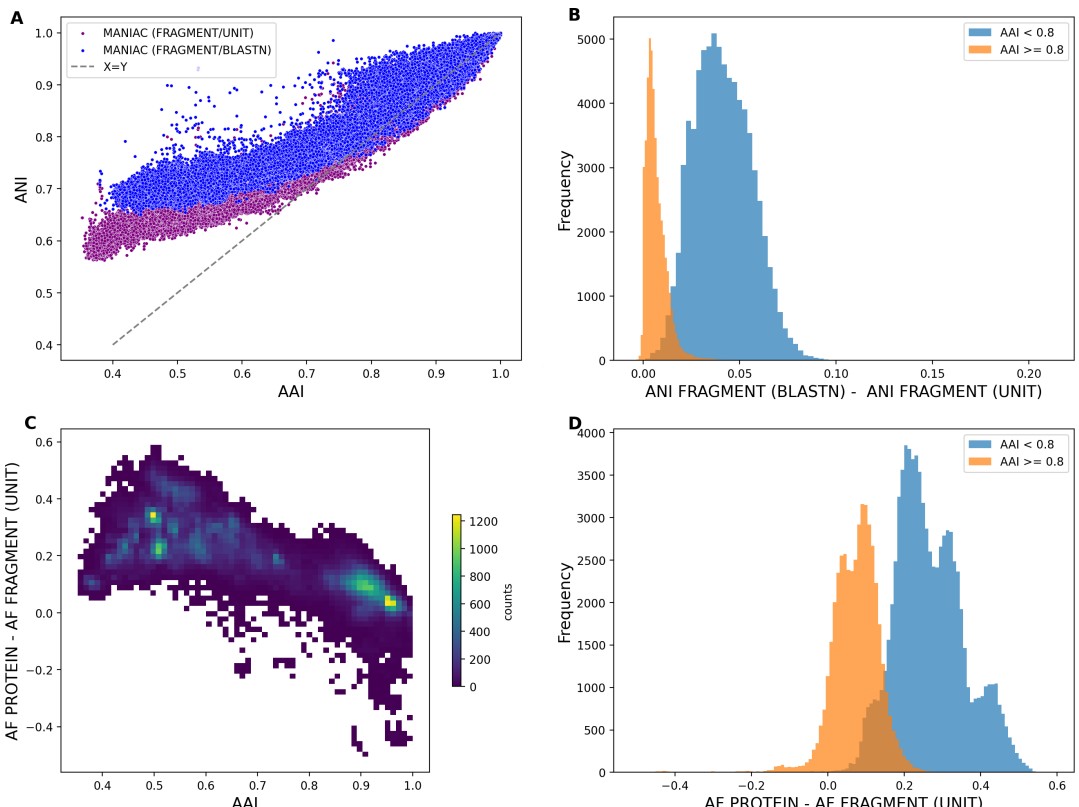

**FIG 4** Comparison of fragment, CDS, and protein modes. (A) ANI (*Y*-axis) vs AAI (*X*-axis) was estimated for the RefSeq data set by MANIAC in the fragment Mode using two scoring schemes: BLASTN (blue) and UNIT (magenta). The dashed line shows the *Y* = *X* relationship. (B) Histogram of ANI difference between BLASTN and UNIT scoring schemes for all pairs with AAI ≥ 0.8 (orange) and AAI < 0.8 (blue). (C) Density plot of differences between AF measured in protein mode and AF measured in fragment mode (*Y*-axis) vs the estimated value of AAI (*X*-axis). (D) Histogram of differences between AF measured in protein mode and AF measured in fragment mode for various ranges of AAI: AAI ≥ 0.8 (orange) and 0.8 > AAI (blue). Only pairs with a minimum alignment length of 5,000 amino acids are shown.

mode (Accurate setting) and AF in the protein mode. Our analysis, depicted in Fig. 5, reveals several observations.

First, our findings align with those of Mavrich and Hatfull, showcasing two evolutionary modes within the phage populations, characterized by a high rate or a low rate of gene flux (36). Importantly, it should be noted that Mavrich and Hatfull employed Mash to estimate nucleotide distance, which does not distinguish well between ANI and AF, particularly for distant genomes. Consequently, their nucleotide distance measurements are influenced by gene content distance, affecting the shape of their plot compared to ours. Despite these methodological differences, the dense clustering of data points in our analysis, indicating high ANI but low AF (see Fig. 5A and B), suggests the presence of many recent horizontal gene transfer (HGT) or recombination events, corroborating the two evolutionary modes identified by Mavrich and Hatfull. Thus, while the approaches differ, the underlying signal from the data is consistent across both studies, reaffirming the identification of these two modes in the evolution of phage populations.

Second, as shown in Fig. 5C and D, the observed ANI distribution is multi-modal, with most pairs having ANI either below 75% or above 90%. A closer comparison with the ANI-AF distribution in Fig. 5A and B suggests that most of these pairs are characterized by AF below 0.25, possibly reflecting rampant genetic mosaicism in dsDNA phages. Hence, to account for this effect, we reexamined the ANI distribution for medium-to-high coverage pairs with AF ⩾ 0.5 only (see Fig. 5C). We saw that while the resulting ANI landscape is rigid, with multiple maxima and minima, there is a clear discontinuity in ANI around 80%: out of 1,345,051 pairs with AF ⩾ 0.5, only 15,349 pairs (around 1.1%) had ANI in the range of 80% and 83%. Furthermore, two of the observed maxima

occur exactly at ANI of 70% and at ANI of 95%. Interestingly, these observations are qualitatively identical when assessing the ANI distribution obtained from the analysis of the GPD data set by MANIAC-Accurate (10M pairs; Fig. S9) and of the PhageScope data set with MANIAC-Fast (240M pairs; Fig. S10). Finally, among the high coverage pairs (AF $\geqslant$ 0.8), we observed few pairs with ANI < 0.9, but for those with high identity, there was a previously observed ANI gap of around 99.5%, concordant with the study by Aldeguer-Riquelme and colleagues (52).

## Assessing the quality of ANI and AF as predictors of same-genus taxonomy

The ANI metric has increasingly become pivotal for standardizing taxonomic classifications (24, 54, 55). In fact, the ICTV's Bacterial Viruses Subcommittee now recommends using ANI cut-offs of 95% for the species rank and 70% for the genus rank across the full genome length (wgANI) for tailed dsDNA phages of the class *Caudoviricetes*. However, to our knowledge, there have been no attempts to systematically examine the genus assignment in bacterial viruses depending on the values of both ANI and AF. Thus, we next examined the predictive quality of these metrics for genomes of bacterial DNA viruses with known taxonomic information.

To carry out this analysis, we created a representative data set consisting of 4,618 species of dsDNA bacterial viruses with the full taxonomic classification provided by ICTV and a corresponding genome sequence (the ICTV data set; see Materials and Methods); of those, 4,581 (99%) belong to the *Caudoviricetes* class. We processed these genomes through MANIAC in all three modes (fragment, CDS, and protein). Figure 6A shows the distribution of ANI and AF (fragment mode) among genome pairs by their genus assignment (same or different), while Fig. 6B shows the distribution of the wgANI depending on the genus assignment (same or different) relative to the ICTV-recommended cutoff of 70%. This analysis shows that, while a strong correlation is evident between genetic relatedness and taxonomic genus classification, the absence of a singular demarcation line shows that taxonomic assignment based on a single, universal ANI/AF cutoff can be misleading. Furthermore, the presence of genome pairs with high ANI yet low AF highlights the prevalence of genetic mosaicism, which could additionally bias the wgANI estimates for more distantly related viral genera in the case of recent genetic exchange.

To evaluate the predictive utility of ANI and AF for genus classification in the statistical sense, we applied a statistical learning approach via logistic regression (see Materials and Methods), treating ANI and/or AF as predictors and genus assignment as the binary outcome (same vs. different). After partitioning the data set into training/validation (65%) and test (35%) sets using the genus level, we conducted 10-fold cross-validation across five models (see Materials and Methods): ANI in fragment mode alone (model 1), ANI + AF in fragment mode (model 2), ANI + AF in CDS mode (model 3), AAI + AF in protein mode (model 4), and ANI in fragment mode + AF in protein mode (model 5). Finally, the performance of these models was assessed by calculating the area under the precision-recall curve on both training/validation and test data sets (PR-AUC) with 95% confidence intervals estimated by performing bootstrap with 100 replicates (see Materials and Methods).

The results, shown in Fig. 6C and Table 3, demonstrate that while the PR-AUC for model 1 (ANI alone) is relatively high, estimated as 0.90, the predictive power for model 2 (ANI + AF) is substantially and significantly higher, estimated as 0.98. This finding emphasizes the criticality of both ANI and AF in taxonomic classification as recommended by ICTV's Bacterial Viruses Subcommittee (30). Second, we hypothesized that models 3–5 which incorporate more information (e.g., via inference of orthology or higher sensitivity of protein alignment) may be more accurate in predicting the taxonomic genus than the fragment mode. Our results did not show any support for this hypothesis, with neither model having a significantly higher value of PR-AUC than model 2 (in fact, in our tests, model 2 significantly outperformed model 3; see Table 3). Thus, contrary to our initial hypothesis, the additional information provided by ORF prediction or protein

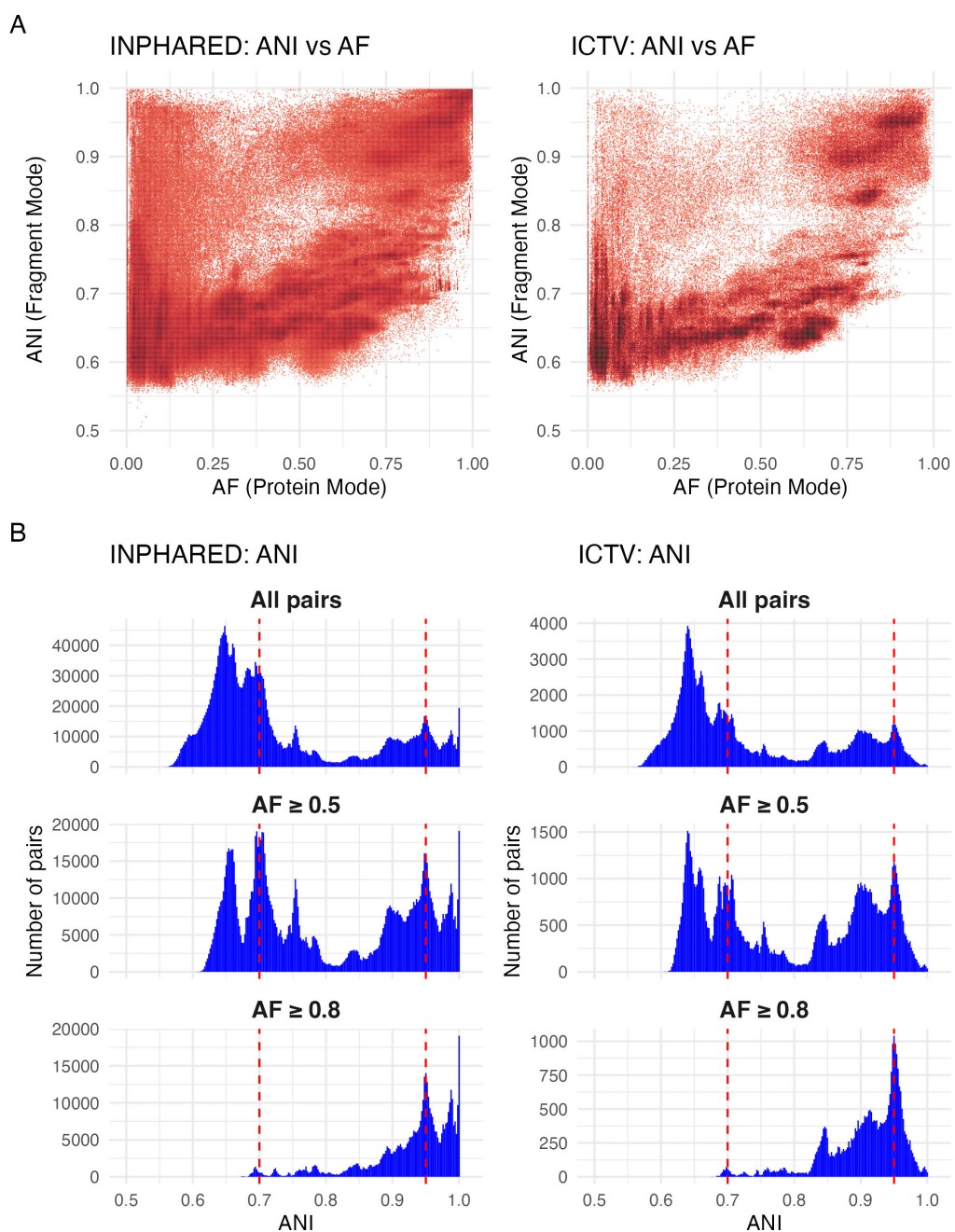

**FIG 5** ANI distribution in bacterial dsDNA viruses. (A) Left: density plot of ANI (*Y*-axis) vs. AF (*X*-axis) for all pairs of 21,754 complete phage genomes from the Inphared data set estimated by MANIAC, with the color intensity reflecting the density. ANI was estimated using the fragment mode and the Accurate parameter setting of MANIAC, while AF was estimated using the protein mode. After filtering (see Materials and Methods), we found 3,868,210 pairs with non-zero values of ANI and AF. Right: same as in the left panel but made for the ICTV data set of 4,618 representative genomes of dsDNA bacterial viruses, showing 234,270 pairs with non-zero values of ANI and AF. (B) Left: histograms of ANI values for genome pairs from the Inphared data set with (top) a minimum alignment length of 2,000 aa; (middle) AF ≥ 0.5 and minimum alignment length of 5,000 aa; and (bottom) AF ≥ 0.8 and minimum alignment length of 5,000 aa. Right: same as left panels but for the ICTV data set. Red dashed lines indicate the ANI values of 70% ANI and 95%.

sequence alignment (CDS and protein modes) did not confer any additional predictive advantage for genus prediction over the straightforward fragment mode calculation of ANI. These results remained qualitatively consistent even when the data were split into

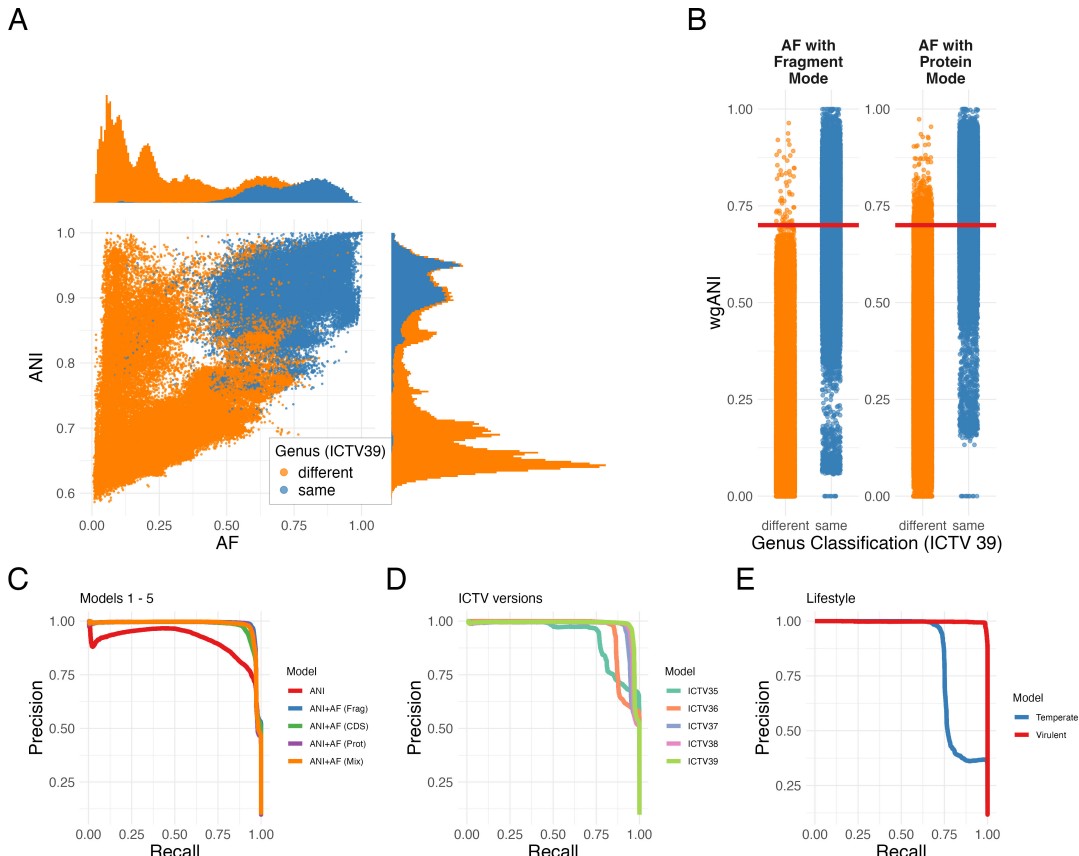

**FIG 6** Impact of ANI and AF on taxonomic same-genus prediction. (A) ANI (*Y*-axis) vs. AF (*X*-axis) for 163,041 pairs from the ICTV data set filtered by a minimum alignment length of 2,000 bp. Points are colored based on genus classification: same genus (blue), different genus (orange). (B) Distribution of wgANI (product of ANI and AF; *Y*-axis) as a function of the genus classification (*X*-axis), with the same color legend as in panel A. The left plot shows the results obtained for AF calculated in the fragment mode, and the right plot shows the results obtained for AF calculated in the protein mode. The red dashed line shows the wgANI of 70%, which is considered a recommended cutoff for same-genus taxonomic assignment in bacterial dsDNA viruses. (C) Precision-recall curves for five models evaluated on the test data set: model 1 (ANI in fragment mode, red), model 2 (ANI + AF in fragment mode, blue), model 3 (ANI + AF in CDS mode, green), model 4 (AAI + AF in protein mode, purple), and model 5 (ANI in fragment mode + AF in protein mode, orange). (D) Precision-recall curves of model 2 across different ICTV releases: 35 (turquoise), 36 (orange), 37 (blue), 38 (pink), and 39 (lime green). (E) Precision-recall curves for model 2, stratified by the predicted phage lifestyle: virulent (red) and temperate (blue). PR-AUC values for curves in panels C, D, and E are provided in Tables 3 to 5, respectively.

training/validation and test data sets based on family assignment rather than genus, despite the reduction of 56% of the original data set (see Fig. S11). Thus, henceforth we continue the analyses for model 2 (ANI + AF in fragment mode) since it offers the best combination of accuracy and simplicity.

The precision and recall values, or the false-detection rate, defined as $1 - \text{precision}$, which are all influenced by the chosen prediction threshold, highlight a trade-off: higher thresholds improve precision (and hence decrease FDR) at the cost of recall, and vice versa. Optimal balance, as indicated by the F1 score, was achieved at a probability threshold of approximately 0.44, which was close to a threshold of 0.47 when using wgANI directly (see Fig. S12). Then, we investigated optimal thresholds for the individual choice of ANI and AF and found that the optimal F1 score is obtained for $\text{ANI} \sim 0.8$ and $\text{AF} \sim 0.3$ (Fig. S13). Regardless of the threshold, however, the model is not perfect and produces false positives even at a high probability threshold and false negatives at a low probability threshold. To investigate those outliers, we looked for pairs of different viral genera predicted to be the same with $P \geqslant 0.95$, as well as for pairs of viruses within the same genus predicted as the same with $P < 0.05$ (or, in other words, predicted as different genera with $P \geqslant 0.95$), using the entire data set for this purpose to

**TABLE 3** PR-AUC estimates with 95% confidence intervals for five principal models obtained on both training/validation data set and test data set

| Data set | Model | PR-AUC | 95% CI |
|---|---|---|---|
| Training/validation | ANI | 0.8903 | (0.8854, 0.8948) |
| Training/validation | ANI + AF (fragment) | 0.9932 | (0.9926, 0.9940) |
| Training/validation | ANI + AF (CDS) | 0.9922 | (0.9913, 0.9930) |
| Training/validation | ANI + AF (protein) | 0.9932 | (0.9925, 0.994) |
| Training/validation | ANI + AF (mixed) | 0.9923 | (0.9914, 0.9931) |
| Test | ANI | 0.9046 | (0.9005, 0.9097) |
| Test | ANI + AF (fragment) | 0.9808 | (0.9789, 0.9827) |
| Test | ANI + AF (CDS) | 0.9748 | (0.9725, 0.9768) |
| Test | ANI + AF (protein) | 0.9781 | (0.976, 0.9802) |
| Test | ANI + AF (mixed) | 0.9780 | (0.9759, 0.9800) |

capture all putative outliers. The first group (false positives) represented 113 out of 1,431 genera (see Fig. S14). The second group (false negatives) consisted of 16 genera and was dominated by three of them: *Fromanvirus* (422 out of 612 pairs), followed by *Phietavirus* (84 pairs) and *Anayavirus* (57 pairs). It is unclear whether these outliers reflect complex taxonomic assignment (e.g., due to genetic mosaicism or extensive genus diversity) or imperfect taxonomic classification in groups like *Fromanviruses* and *Phietaviruses;* however, it is worth noting that 508 of the 612 pairs were significantly assigned to the temperate lifestyle, while only 1 pair was significantly assigned to the virulent lifestyle. To display all possible false positive predictions, we generated a network of false-positive genera from the entire ICTV data set and assigned them to putative families or subfamilies to facilitate potential verification in the future (Fig. S14).

We next compared the performance of model 2 (trained on ICTV39) on previous different ICTV releases, hypothesizing that the genus prediction accuracy may improve with subsequent releases. Indeed, as shown in Fig. 6D and Table 4, we found the model's improved performance between subsequent releases 35, 36, 37, and 38, showing a statistically significant increase in PR-AUC when predicting the same genus on every subsequent ICTV release. While this observation is likely partly influenced by the increasing size of the VMR data sets with every release, the main enhancements likely originate from the ICTV's continuous refinements and updates in taxonomic classification, which take ANI and AF into consideration.

Finally, we investigated the performance of model 2 depending on the phage lifestyle (temperate or virulent). We found that virulent phage pairs yielded significantly stronger predictive outcomes of model 2 (PR-AUC of 0.997) compared to temperate pairs (PR-AUC of 0.847), as illustrated in Fig. 6E and Table 5. This variance may not only reflect the compositional bias of our data set—temperate phages being less represented—but may also hint at the intrinsic complexity of taxonomic classification amongst temperate phages. The latter are more likely to undergo horizontal gene transfer (HGT) or recombination than virulent phages, leading to increased genetic mosaicism (56, 57). Combining this information with the false-positive data set, where the majority of unique genera were assigned to the temperate lifestyle (Fig. S12), we can clearly observe genera of temperate phages that belong to the same subfamily cluster together, such as members of the subfamilies *Hendrixvirinae, Munstervirinae, Sepvirinae, Azeredovirinae,*

**TABLE 4** PR-AUC estimates with 95% confidence intervals for model 2 on different ICTV releases[a]

| ICTV version | PR-AUC | 95% CI |
|---|---|---|
| ICTV 39 | 0.9808 | (0.9789, 0.9827) |
| ICTV 38 | 0.9749 | (0.9728, 0.9774) |
| ICTV 37 | 0.9687 | (0.9656, 0.9718) |
| ICTV 36 | 0.9472 | (0.9437, 0.9509) |
| ICTV 35 | 0.9296 | (0.925, 0.9348) |

[a]Results were obtained on the test data set.

**TABLE 5** PR-AUC estimates with 95% confidence intervals for model 2 on different phage lifestyles[a]

| Lifestyle | PR-AUC | 95% CI |
|-----------|--------|--------|
| Virulent | 0.9970 | (0.9958, 0.9979) |
| Temperate | 0.8465 | (0.8342, 0.8586) |

[a]Results were obtained on the test data set.

and genera related to the genus *Lambdavirus*. Mosaicism in the "Lambda supercluster" has been well documented, with different genus pairs sharing different sections of the genome (58). This module shuffling makes it more challenging to define clear taxonomic boundaries to use as demarcation criteria and poses a sophisticated challenge in using ANI and AF to predict genus classification. Selecting a set of hallmark genes as the unit of evolutionary information is the current gold standard in virus taxonomy to overcome this issue (30, 55).

## DISCUSSION

In this study, we introduced MANIAC, a novel computational pipeline designed to address an important gap in the field by enabling accurate and efficient calculation of ANI and AF for divergent viral genome pairs, while scaling effectively to data sets comprising tens or hundreds of thousands of genomes. MANIAC demonstrates a near-perfect correlation with the BLAST-based ANIb in estimating ANI and AF at a fraction of its runtime. When compared to other widely used methods such as MUMmer, FastANI, CheckV, and skani, MANIAC consistently outperformed these methods in accurately estimating ANI for genomes with divergence near 70%, a range where existing methods often struggle. Simulated data sets were employed to optimize MANIAC's parameters, enabling accurate predictions of genetic divergence down to ANI values as low as 60%–70%, a finding corroborated by real genomic data and validated through comparisons with average AAI. Furthermore, by using MANIAC's more efficient parameterization (cf., Table 1), it is possible to compromise slightly on accuracy (especially of AF estimates) while still outperforming other methods in ANI estimation for divergent genomes, achieving run times comparable to FastANI and CheckV. These results highlight MANIAC's dual capability of precision and scalability, making it a valuable method for addressing the challenges of large-scale viral genome comparisons and improving taxonomic assignments in virology.

One important application of having an accurate and scalable method to estimate ANI and AF was to investigate the existence of the "ANI gap" in populations of bacterial dsDNA viruses. While the phenomenon of the ANI gap has been extensively studied in bacteria, much less attention has been given to its occurrence in viruses. One reason for this disparity is the historical lack of methods capable of analyzing ANI distributions across millions of pairwise comparisons and a broad range of ANI values. Our findings highlight a complex ANI landscape in bacterial dsDNA viruses, characterized by multiple hills and valleys. This complexity likely reflects high mutation rates, absence of universal genetic markers, frequent recombination events, reliance on host cells and diverse host ranges, all of which blur traditional species boundaries (50). In spite of the rigidness, we observed a distinct gap around an ANI of 80%, with only 1% of pairs falling within the range of ANI 80%–83%. These findings are broadly consistent with two previous studies that identified a gap in ANI around 85–90% in large-scale viral genome comparisons (51). While this range is slightly higher than the gap observed in our study, it is important to note that the two previous analyses included substantial numbers of non-bacterial viruses and utilised tools such as MUMmer or FastANI, which our benchmarking suggests tend to overestimate ANI below 80% (cf. Fig. 2). In contrast, the only study to date that we know of that specifically examined ANI distribution in bacterial dsDNA viruses, by Accetto and Janež (53), reported a comparable ANI gap, but their analysis was based on a limited dataset of 186 lytic myoviruses infecting *Enterobacteriaceae*. Furthermore, the ANI distributions derived from metagenome-assembled phages (Fig. S9 and S10) also show

quantitative differences, indicating that the precise range of the ANI gap may depend on the composition of the underlying population, as previously suggested (50).

Regardless of its precise boundaries, the ANI gap in bacterial dsDNA viruses bears resemblance to the well-documented ANI gap in bacteria, the latter typically observed around an ANI of 90% (7, 23) Various studies have pointed to recombination being one of the key mechanisms that contributes to the maintenance of the ANI gap. Specifically, almost all bacterial species exhibit recombination to varying degrees (59, 60). Recombination helps maintain genetic homogeneity up to a certain level of genetic divergence, beyond which it shifts to accelerate divergence (61, 62). In line with this notion, bacterial genomes with an ANI of around 90% are known to exhibit unusually low rates of gene flow (63), which, in turn, is consistent with the observation that homologous recombination requires high DNA sequence identity—typically above 90% —to enable recombinase-mediated repair. The discontinuity in viral ANI we observe may be driven by similar evolutionary dynamics, and the shift in the ANI gap from 90% ANI in bacteria to approximately 80% in their viruses could reflect unique adaptations in the latter. Bacterial viruses are known to often encode diverse recombinase systems that are more tolerant of genetic divergence within homologous DNA fragments (64– 66). This raises the intriguing possibility that, as in bacteria, there exists a threshold of genetic divergence in bacterial dsDNA viruses (on average around ANI 80%) above which recombination helps maintain population cohesiveness, akin to its role in bacteria (67, 68). Conversely, below this threshold recombination becomes a major diversifying force, resulting not only in lower ANI values (greater genetic diversity) but also in greater genomic mosaicism as evidenced by reduced AF.

A related question then is how the extensive genetic diversity and mosaicism impact taxonomic classification at the genus level in bacteria dsDNA viruses. To address this question, we applied a machine learning approach to assess how well various metrics of ANI and AF impact the prediction of same-genus taxonomic classification. As expected, our results demonstrate that ANI and AF are excellent predictors of the same-genus assignment, achieving a 99% precision with the recall of 92% or 95% precision with the recall of 96% with the ICTV39 release (VMR_MSL39_v2). Investigation of false-positive predictions reveals that, at the level of 95% probability, 69% of the identified genome pairs (969 out of 1,403) belong to the *Demerecviridae* family. The false-negative predictions in turn were mostly detected for the *Fromanvirus*, *Phietavirus,* and *Anayavirus* genera, all of which were predicted to have a temperate lifestyle. In fact, when we fitted the best model to only viruses with a predicted virulent lifestyle, the model fitted significantly better (PR-AUC increased from 0.981 to 0.997, and recall of 99% was achieved for a precision of 99%). However, the model's predictive power decreased significantly for temperate viruses (PR-AUC of 0.847). The implication that taxonomic classification may be more challenging in temperate phages can be expected due to the fact that temperate phages are known to undergo more extensive recombination than lytic phages and hence are expected to demonstrate higher levels of genetic mosaicism (56, 67). Therefore, while ANI and AF are clearly very strong predictors of the taxonomic genus, such taxonomic assignment in big data should be carried out with caution, especially if the viruses are temperate.

MANIAC builds on the fragment-based approach originally proposed by Goris et al. for bacterial genomes, in which a query genome is chopped into fragments and ANI is calculated as the weighted percentage identity of aligned fragments (1). This methodology contrasts with whole-genome alignment methods used by tools such as CheckV, which calculate ANI and AF based on the alignment of full, unfragmented genomes. The fragment-based approach offers several key advantages: by aligning genome fragments independently, it ensures homologous regions are captured without being overshadowed by non-alignable or highly divergent regions that could cause the full-genome alignment to fail. In addition, it provides a more unbiased estimate of ANI by mitigating the effects of genetic rearrangements such as inversions or translocations, which can mislead whole-genome alignments. Adjusting scoring schemes to detect lower degrees

of similarity further complicates whole-genome alignment by resulting in fragmented alignments with variable E-values, making consistent ANI calculations challenging. By chopping genomes into fragments, MANIAC simplifies this process by selecting the best hit for each fragment, ensuring a clearer, more accurate measure of genetic related-ness–especially for viruses with ANI around 70%. However, for closely related genomes with minimal genomic rearrangements and more uniform ANI across their genomes, the fragment-based approach can introduce limitations. Specifically, homologous regions may be split differently across genomes due to differential fragmentation, leading to alignment failures for some fragments. Despite this drawback in such cases, MANIAC was developed to prioritize accurate and efficient ANI estimation in divergent viral genomes. While tools like FastANI and CheckV are highly appropriate for identifying close relationships around 95% ANI, the fragment-based approach used by MANIAC proves advantageous for a more reliable estimation of genetic relatedness in divergent viral genomes.

In conclusion, we argue that MANIAC is a valuable addition in the field of genomic analysis of viruses, providing a reliable framework for ANI estimation of around 70%, as evidenced here on a data set of bacterial dsDNA viruses. On a laptop with an Apple M3 chip with 11 cores, 36 GB of RAM memory, and SSD hard drive, running MANIAC in the fragment mode on the RefSeq data set (4,655 genomes) took us 4 hours and 25 minutes in the "Accurate" parameter setting and 12 minutes and 50 seconds in the "Fast" parameter setting. Hence, we expect that MANIAC to become useful in large-scale viral genomics aimed at capturing and quantifying the extensive diversity of viruses belonging to different operational taxonomic units.

## ACKNOWLEDGMENTS

This work was financed by the Polish National Agency for Academic Exchange (J.K. and R.J.M.), the Polish National Science Centre OPUS grant (grant 2020/37/B/NZ8/03492; W.N., J.K. and R.J.M.), the EMBO Installation grant (W.N., J.L., J.K. and R.M.), the Polish National Science Centre Sonata Bis grant (grant 2020/38/E/NZ8/00432; J.L. and R.J.M.) and the International Visegrad Fund scholarship (J.H.). L.C. acknowledges funding from the MRC Centre for Global Infectious Disease Analysis (reference MR/X020258/1), funded by the UK Medical Research Council (MRC). This UK-funded award is carried out in the frame of the Global Health EDCTP3 Joint Undertaking. E.M.A. gratefully acknowledges funding by the Biotechnology and Biological Sciences Research Council (BBSRC); this research was funded by the BBSRC Institute Strategic Programme Microbes and Food Safety BB/X011011/1 and its constituent projects, BBS/E/F/000PR13634, BBS/E/F/000PR13635 and BBS/E/F/000PR13636, and the BBSRC Institute Strategic Programme Food Microbiome and Health BB/X011054/1 and its constituent project BBS/E/F/000PR13631. We gratefully acknowledge Polish high-performance computing infrastructure PLGrid (HPC Centers: CI TASK, ACK Cyfronet AGH) for providing computer facilities and support within computational grant nos. PLG/2023/016857 and PLG/2024/017509.

We would also like to acknowledge Bogna Smug for her valuable input and insightful comments provided at various stages of the research and manuscript preparation.

## AUTHOR AFFILIATIONS

[1]Malopolska Centre of Biotechnology, Jagiellonian University, Kraków, Poland
[2]Doctoral School of Exact and Natural Sciences, Jagiellonian University, Kraków, Poland
[3]Faculty of Biochemistry, Biophysics and Biotechnology, Jagiellonian University, Kraków, Poland
[4]Department of Infectious Disease Epidemiology, School of Public Health, Imperial College London, London, United Kingdom
[5]Quadram Institute Bioscience, Norwich Research Park, Norwich, United Kingdom

## AUTHOR ORCIDs

Evelien M. Adriaenssens  http://orcid.org/0000-0003-4826-5406
Rafal J. Mostowy  http://orcid.org/0000-0002-4557-3748

## FUNDING

| Funder | Grant(s) | Author(s) |
|---|---|---|
| Narodowe Centrum Nauki (NCN) | 2020/37/B/NZ8/03492 | Wanangwa Ndovie |
| | | Rafal J. Mostowy |
| Narodowa Agencja Wymiany Akademickiej (NAWA) | PPN/PPO/2018/1/00021 | Janusz Koszucki |
| | | Rafal J. Mostowy |
| Narodowe Centrum Nauki (NCN) | 2020/38/E/NZ8/00432 | Jade Leconte |
| | | Rafal J. Mostowy |
| UKRI \| Medical Research Council (MRC) | MR/X020258/1 | Leonid Chindelevitch |
| UKRI \| Biotechnology and Biological Sciences Research Council (BBSRC) | BB/X011011/1, BBS/E/F/000PR13634, BBS/E/F/000PR13635, BB/X011054/1, BBS/E/F/000PR13631 | Evelien M. Adriaenssens |
| European Molecular Biology Organization (EMBO) | Installation Grant to Rafal Mostowy | Wanangwa Ndovie |
| | | Jade Leconte |
| | | Janusz Koszucki |
| | | Rafal J. Mostowy |
| International Visegrad Fund (IVF) | | Jan Havránek |

## AUTHOR CONTRIBUTIONS

Wanangwa Ndovie, Data curation, Formal analysis, Investigation, Methodology, Visualization, Writing – review and editing | Jan Havránek, Conceptualization, Formal analysis, Investigation, Methodology, Visualization, Writing – original draft, Software | Jade Leconte, Data curation, Software, Validation, Writing – review and editing | Janusz Koszucki, Software, Validation, Writing – review and editing | Leonid Chindelevitch, Investigation, Methodology, Writing – review and editing, Funding acquisition | Evelien M. Adriaenssens, Formal analysis, Investigation, Writing – review and editing, Funding acquisition | Rafal J. Mostowy, Conceptualization, Data curation, Formal analysis, Funding acquisition, Investigation, Methodology, Resources, Supervision, Validation, Writing – original draft, Writing – review and editing, Project administration, Visualization

## ADDITIONAL FILES

The following material is available online.

### Supplemental Material

**Supplemental Figures (mSystems01661-24-s0001.pdf).** Figures S1 to S12.
**Supplemental Tables (mSystems01661-24-s0002.xlsx).** Tables S1 to S4.

### Open Peer Review

**PEER REVIEW HISTORY (review-history.pdf).** An accounting of the reviewer comments and feedback.

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
