## [Reviewer comments · mSystems]

Exploration of the genetic landscape of bacterial dsDNA viruses reveals an ANI gap amidst extensive mosaicism

Wanangwa Ndovie, Jan Havránek, Jade Leconte, Janusz Koszucki, Leonid Chindelevitch, Evelien Adriaenssens, and Rafal Mostowy

Corresponding Author(s): Rafal Mostowy, Jagiellonian University in Krakow

Review Timeline:

Submission Date:

December 10, 2024

Accepted:

January 6, 2025

Editor: Juliette Hayer

Reviewer(s): The reviewers have opted to remain anonymous.

Transaction Report:

DOI: <https://doi.org/10.1128/msystems.01661-24>

Re: mSystems01661-24 (Exploration of the genetic landscape of bacterial dsDNA viruses reveals an ANI gap amidst extensive mosaicism)

Dear Dr. Rafal Mostowy:

Regarding the highly improved version of the manuscript, and considering that you have addressed all the comments from the reviewers, I am happy to accept this manuscript for publication.

Therefore, Reviewer #1 had some additional remarks regarding Figure 5, so I leave you check this and fix it if needed for the final version.

Your manuscript has been accepted, and I am forwarding it to the ASM production staff for publication. Your paper will first be checked to make sure all elements meet the technical requirements. ASM staff will contact you if anything needs to be revised before copyediting and production can begin. Otherwise, you will be notified when your proofs are ready to be viewed.

Cover Image Submissions: If you would like to submit a potential Cover Image, please email a file and a short legend to mssystems@asmusa.org. Please note that we can only consider images that (i) the authors created or own and (ii) have not been previously published. By submitting, you agree that the image can be used under the same terms as the published article. Image File requirements: TIF/EPS, 7.5 inches wide by 8.25 inches tall (at least 2,250 pixels wide by 2,475 pixels tall), minimum 300 dpi resolution (600 dpi preferred), RGB, and no figure elements, e.g., arrows or panel labels. The legend should be a short description of the image, 1-2 sentences recommended. Please download and use this interactive template in Adobe to ensure that your proposed cover image meets our size requirements (<https://journals.asm.org/pb-assets/pdf-text-excel-files/ASM-Interactive-Sizing-Cover-Template-1715689791.pdf>).

We recognize that the video files can become quite large, so to avoid quality loss ASM suggests sending the video file via <https://www.wetransfer.com/>. When you have a final version of the video and the still ready to share, please send it to mSystems

staff at mSystems@asmusa.org.

Sincerely,
Juliette Hayer
Editor
mSystems

Reviewer #1 (Comments for the Author):

The revised article by Ndovie and colleagues represents a much-improved version, and the authors have made a sincere effort to address my comments/suggestions and those of the other reviewers. I want to thank the authors for their efforts and comprehensive revisions. I have no more specific suggestions for the authors to consider except one, on the big picture. I don't know what else the authors could do about this one, however, so I leave it completely to their discretion to do more or do nothing about it.

Their ANI plots (e.g. Figure 5) do not look as clean in terms of the gap as those presented in previous publications. For instance, Aldeguer et al., (mBio 2024) show a clearer gap around 95% ANI than observed here and I believe, these authors are using some of the exact same genomes but probably many additional genomes (see their supplementary figures). Similarly for the figure 3 from Roux et al., Nat. Biotechnol. 2019 with fewer genomes included. I think there is likely somewhere a problem/issue (but I could be wrong about this). It could be more noisy, higher error genomes used here or the actual processing of the match results (i.e. the MANIAC code), but I am not sure. That's the main reason that I don't want to insist more on this and I leave it to the discretion of the authors to check this issue further. A few things that the authors could possibly check are listed below.

The authors propose 3 different settings called "ANlb", "Accurate" and "Fast". In Table 2, they calculate the correlation between MANIAC and ANlb with the "ANlb parameter set" and this looks great and robust. So, I would believe results in terms of the gap based on the ANlb set. However, in the key Figure 5 that I am talking about, the authors use the "Accurate" parameters. Could this introduce more noisy ANI value distribution, e.g., maybe shift lower ANI values up/higher?

The authors only considered cultured phage genomes isolated from a few bacterial taxa; I think Aldeguer et al and Roux et al., used many more genomes and taxa. Could this account for a difference in the ANI value distributions and how pronounced (or not) the ANI gap observed may be?

Finally, in the legend of Figure 5, the authors say "ANI was estimated using the Fragment Mode and the Accurate parameter setting of MANIAC, while AF was estimated using the Protein Mode." So, if I get this right, the authors basically mixed the nucleotide identity and the amino acid alignment to filter matches. Why would you do this? I think it would make more sense to use the alignment fraction calculated with the nucleotide alignment. Could this introduce some biased ANI values somewhat e.g. shift them upwards? I am not sure.

Reviewer #2 (Comments for the Author):

I thank the authors for addressing all the previous remarks from reviewers, and have no further comments at this stage